# Robust Detection Outcome:
# A Metric for Pathology Detection in Medical Images

**Felix Meissen**[*1]                                                       FELIX.MEISSEN@TUM.DE
[1] *Technical University of Munich*

**Philip Müller**[*1]                                                    PHILIP.J.MUELLER@TUM.DE
**Georgios Kaissis**[1,2]                                                       G.KAISSIS@TUM.DE
[2] *Helmholtz Zentrum Munich*

**Daniel Rueckert**[1,3]                                                 DANIEL.RUECKERT@TUM.DE
[3] *Imperial College London*

**Editors:** Accepted for publication at MIDL 2023

## Abstract

Detection of pathologies is a fundamental task in medical imaging and the evaluation of algorithms that can perform this task automatically is crucial. However, current object detection metrics for natural images do not reflect the specific clinical requirements in pathology detection sufficiently. To tackle this problem, we propose *Robust Detection Outcome* (RoDeO); a novel metric for evaluating algorithms for pathology detection in medical images, especially in chest X-rays. RoDeO evaluates different errors directly and individually, and reflects clinical needs better than current metrics. Extensive evaluation on the ChestX-ray8 dataset shows the superiority of our metrics compared to existing ones. We released the code at https://github.com/FeliMe/RoDeO and published RoDeO as pip package (rodeometric).

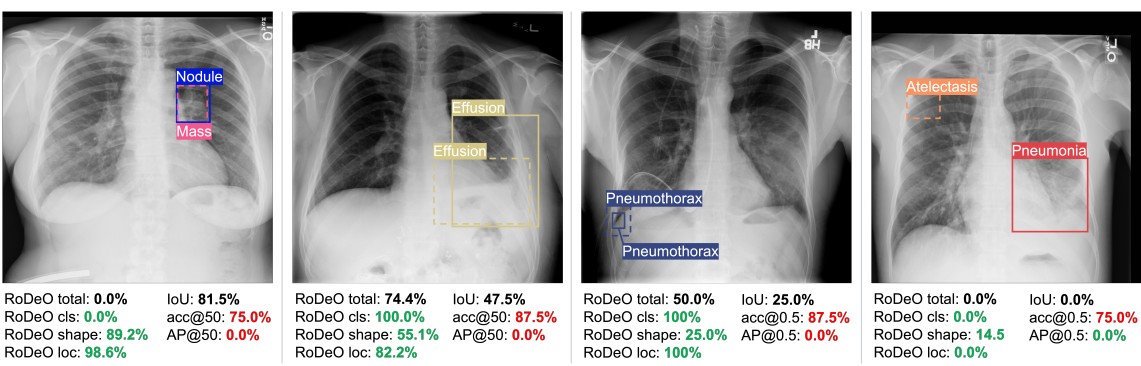

Figure 1: Example images from the ChestX-ray8 dataset (Wang et al., 2017) with example predicted (dashed) and target (solid) bounding boxes, and the corresponding IoU, acc@IoU, AP@IoU, and RoDeO results. **Left to right**: Correct box with misclassification, partly overlapping box, too large predicted box, wrong prediction.

**Keywords:** Metric, Pathology Detection, Object Detection.

---

* Contributed equally

## 1. Introduction

Localization of pathologies in medical images is of high clinical relevance. It does not only speed up diagnosis but is also valuable to asses the interpretability of machine learning models. Bounding boxes are especially useful in this regard, as they both satisfy the clinical need for coarse localization and are easier to obtain than exact segmentations. However, assessing the quality of predicted bounding boxes is non-trivial. For object detection on natural images, metrics based on IoU-thresholds – like *Average Precision at IoU (AP@IoU)* or *mean Average Precision (mAP)* – are typically used. We found these to be unsuitable for pathology detection in some medical images, such as chest radiographs.

The metric *AP@IoU* first performs a greedy matching between predicted and target boxes while only considering predicted boxes that have the same class and an *Intersection over Union (IoU)* above a defined threshold with the target box. Each target box can be matched to at most one predicted box (the one with the highest IoU). All unmatched boxes and those below the threshold are considered false positives. In the next step, the *average precision (AP)* between the associated classes of the predicted and target boxes is computed. To obtain the mAP, the results of multiple IoU thresholds are averaged. Similarly, other classification metrics can be converted to detection metrics by computing them at IoU-thresholds, such as acc@IoU, recall@IoU, or precision@IoU (cf. Appendix E).

Although the tasks are similar, the requirements of object detection in natural images and pathology detection in medical images are much different. For medical pathology detection, coarse localization is already clinically useful and the detection of pathologies is valuable even under misclassification since radiologists are well-trained to classify a pathology correctly once it is detected. Lastly, different sources of detection errors need to be measured separately, a common requirement for all object detection tasks. Current object detection metrics do not reflect these requirements well: IoU-based metrics are commonly used at high IoU-thresholds ($\geq 0.5$). Since these are typically hard to obtain in pathology detection, and earlier works have used significantly lower thresholds (Baumgartner et al., 2021; Jaeger et al., 2020). Additionally, these metrics do not differentiate between different types of errors: misclassification, faulty localization, box shape mismatch, and overprediction all cause false positives and thus degrade the score in the same way, although their underlying sources are much different. This makes metrics like AP@IoU hard to interpret and models hard to compare if they are only evaluated under this metric. Moreover, AP@IoU and mAP are sensitive to misclassification and do not attribute the detection of pathologies if their predicted class is wrong. Lastly, all metrics based on IoU thresholds only have a limited notion of proximity and drop harshly if the box overlap falls below the IoU-threshold. Figure 1 shows cases where AP@IoU (and therefore also mAP) and acc@IoU exhibit such undesired behaviors.

To tackle the aforementioned weaknesses of existing metrics, we introduce *Robust Detection Outcome (RoDeO)*, a new metric for pathology detection based on three types of errors that cause the detection quality to deteriorate: classification errors, localization errors, and shape mismatch. RoDeO is easy to interpret, returns sub-scores for different error types, degrades gracefully, and reflects the specific requirements of pathology detection.

**Our contributions are the following**

- We identify weaknesses in current detection metrics when used for pathology detection.

- We propose RoDeO, a new metric for pathology detection that better reflects the clinical needs for course localization and separation of different error sources.

- We perform extensive evaluation of our proposed metric compared to existing ones and show that RoDeO is then only metrics capable of identifying weaknesses of an exemplary pathology detection model.

## 2. Related Work

The default metric for object detection in computer vision tasks is average precision. In the PASCAL VOC challenge (Everingham et al., a,b), AP@50 is used, while in the COCO detection challenge (Lin et al., 2014), mAP is the main challenge metric (with IoU-thresholds from 0.5 to 0.95 in steps of 0.05). Several attempts have been made to isolate different error types of AP@IoU and mAP by observing how the metric changes when certain errors in the predictions are fixed (Bolya et al., 2020; Borji and Iranmanesh, 2019; Hoiem et al., 2012). However, these techniques are merely fighting symptoms of average precision and inherit all other weaknesses of the metric minus its explainability. Moreover, they add another layer of complexity on top of the original metrics, again complicating their interpretability. Recall@IoU – or specifically the mean average recall over multiple IoU thresholds – is also evaluated in the COCO detection challenge. Other popular object detection metrics are also IoU-based. To evaluate a baseline model on the ChestX-ray8 dataset, Wang et al. (2017) use the accuracy or number of average positive predictions at a given threshold [1].

All of the above metrics compute classification quantities after thresholding at an IoU. RoDeO overcomes this limitation and provides per-error metrics that are easily interpretable, degrade gracefully, and are tailored to clinical needs in pathology detection.

Recently, Maier-Hein et al. (2022) stressed the importance stressed the importance of adequate metric selection for the usefulness of machine learning algorithms in clinical practice. While they acknowleddge that rough localization lies in the nature of object detection tasks, they only consider threshold-based metrics, such as AP Our proposed metric follows their recommendation in first solving the assignment issue and then computing the selected metric on top, but we complement sole classification with other important factors, such as localization and shape correspondence.

## 3. RoDeO Metric

**Overview**  In RoDeO, firstly one-to-one correspondences between the predicted and target bounding boxes are established. Then, three sub-metrics (localization, shape similarity, and classification) are computed using the matched boxes. The scores for over- and underpredictions are then linearly combined with the scores of each sub-metric. Lastly, the harmonic mean of the three sub-metrics is computed to obtain one summary metric.

**Matching**  To be able to compute subsequent sub-metrics, first, a matching between the predicted and target boxes is required for every image. We find these correspondences using the Hungarian method, taking into account correct classifications, as well as shape

---

1. Wang *et al.* chose the Intersection over Bounding Box (IoBB) instead of the IoU. This metric does not consider the size of the predicted box in the denominator and therefore favors large boxes.

and distance information via the generalized IoU (gIoU) (Rezatofighi et al., 2019). Given the sets of target and predicted boxes, $\mathcal{M}^{\mathfrak{t}}$ and $\mathcal{M}^{\mathfrak{p}}$, the cost matrix $\mathcal{C}$ is:

$$\mathcal{C}_{\mathcal{B}^{\mathfrak{t}},\mathcal{B}^{\mathfrak{p}}} = -\mathbb{1}_{[y_{\mathcal{B}^{\mathfrak{t}}}=y_{\mathcal{B}^{\mathfrak{p}}}]} * \omega_{cls} - \mathrm{gIoU}(\mathcal{B}^{\mathfrak{t}},\mathcal{B}^{\mathfrak{p}}) * \omega_{\mathrm{shape}} \quad \forall \mathcal{B}^{\mathfrak{t}} \in \mathcal{M}^{\mathfrak{t}}, \mathcal{B}^{\mathfrak{p}} \in \mathcal{M}^{\mathfrak{p}}, \tag{1}$$

with $\mathcal{B}^{\mathfrak{t}}$ and $\mathcal{B}^{\mathfrak{p}}$ being the boxes – described by a vector $(x,y,w,h)^T$ – and $y_{\mathcal{B}^{\mathfrak{t}}}$ and $y_{\mathcal{B}^{\mathfrak{p}}}$ being the classes of the target and predicted box respectively. $\omega_{\mathrm{shape}}$ and $\omega_{\mathrm{cls}}$ are parameters to weigh the importance of the two terms. By default, $\omega_{\mathrm{shape}}$ is set to 1, and $\omega_{\mathrm{cls}}$ is determined by the sample-wise classification performance of the predictions such that wrong class information does not confuse the matching when the model's class predictions are inaccurate. We thus set $\omega_{\mathrm{cls}} = \max\left(0, \mathrm{MCC}\left(\boldsymbol{y}^{\mathfrak{t}}, \boldsymbol{y}^{\mathfrak{p}}\right)\right)$, where $y_i^{\mathfrak{t}}$ and $y_i^{\mathfrak{p}}$ are the classes of all target and predicted bounding boxes for image $i$, respectively, and MCC is Matthews Correlation Coefficient (Matthews, 1975), as defined in Eq. (14) in Appendix E. This results in a set of matched boxes denoted by $\mathcal{M} = (\mathcal{B}^{\mathfrak{p}}, \mathcal{B}^{\mathfrak{t}}) \mid \mathcal{B}^{\mathfrak{p}} \in \mathcal{M}^{\mathfrak{p}}, \mathcal{B}^{\mathfrak{t}} \in \mathcal{M}^{\mathfrak{t}}$, where $(\mathcal{B}^{\mathfrak{p}}, \mathcal{B}^{\mathfrak{t}})$ is a pair of matched bounding boxes. The set of unmatched targets is denoted by $\mathcal{U}^{\mathfrak{t}} = \mathcal{M}^{\mathfrak{t}} \setminus \mathcal{B}^{\mathfrak{t}} \mid (\mathcal{B}^{\mathfrak{p}}, \mathcal{B}^{\mathfrak{t}}) \in \mathcal{M}$, and the set of unmatched predicted bounding boxes is denoted by $\mathcal{U}^{\mathfrak{p}} = \mathcal{M}^{\mathfrak{p}} \setminus \mathcal{B}^{\mathfrak{p}} \quad \forall (\mathcal{B}^{\mathfrak{p}}, \mathcal{B}^{\mathfrak{t}}) \in \mathcal{M}$.

**Localization** We measure the localization quality of the matched boxes in $\mathcal{M}$ by

$$\mathrm{RoDeO}_{\mathrm{matched}}^{\mathrm{loc}} = \frac{1}{|\mathcal{M}|} \sum_{(\mathcal{B}^{\mathfrak{t}},\mathcal{B}^{\mathfrak{p}})\in\mathcal{M}} \exp\left[-\frac{\left(\frac{\mathcal{B}_x^{\mathfrak{t}}-\mathcal{B}_x^{\mathfrak{p}}}{\mathcal{B}_w^{\mathfrak{t}}}\right)^2 + \left(\frac{\mathcal{B}_y^{\mathfrak{t}}-\mathcal{B}_y^{\mathfrak{p}}}{\mathcal{B}_h^{\mathfrak{t}}}\right)^2}{C}\right], \tag{2}$$

where $C$ is a scaling coefficient, set such that for a box pair $(\mathcal{B}^{\mathfrak{t}}, \mathcal{B}^{\mathfrak{p}})$ with relative Euclidean distance equalling one, the value is exactly 0.5, i.e. $\frac{1}{C} = \ln 2 \approx 0.69$. The localization value for a single box pair thus follows the probability density of a (2D separable) normal distribution, such that (i) its maximum value is reached when box centers are aligned, (ii) it degrades slowly for small distances, then fast, until degrading slowly again for large distances. Compared to IoU-based distance functions (used by acc@IoU, AP@IoU, or mAP), which degrade to zero very quickly and prefer axis-aligned boxes; $\mathrm{RoDeO}_{\mathrm{matched}}^{\mathrm{loc}}$ degrades smoothly and isotropically in euclidean space, as highlighted in Figure 2.

**Shape** The shape score encodes differences in size and aspect ratio but must be independent of the distance between the boxes as this is already measured in the localization score. We compute it as the Intersection over Union (IoU) between each matched predicted and target box when assuming the boxes have the same center coordinates (cIoU):

$$\mathrm{RoDeO}_{\mathrm{matched}}^{\mathrm{shape}} = \frac{1}{|\mathcal{M}|} \sum_{(\mathcal{B}^{\mathfrak{t}},\mathcal{B}^{\mathfrak{p}})\in\mathcal{M}} \mathrm{cIoU}(\mathcal{B}^{\mathfrak{t}},\mathcal{B}^{\mathfrak{p}}). \tag{3}$$

We refer to Appendix C for discussion on the preference of cIoU over alternative options.

**Classification** As classification score, RoDeO again uses MCC (c.f. Equation (14) in Appendix E):

$$\mathrm{RoDeO}_{\mathrm{matched}}^{\mathrm{cls}} = \max\left(0, \mathrm{MCC}(\mathcal{Y}^{\mathfrak{t}}, \mathcal{Y}^{\mathfrak{p}})\right), \tag{4}$$

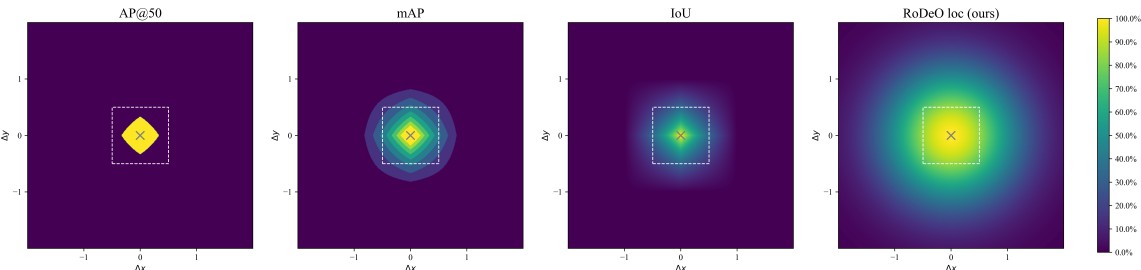

Figure 2: Sensitivity of different metrics to position offsets. For simplicity, we assume both, the target and predicted box, to be square and equal-sized. We plot the values of each metric with position offsets ($\Delta x = \frac{\mathcal{B}_x^{\text{t}} - \mathcal{B}_x^{\text{p}}}{\mathcal{B}_w^{\text{t}}}, \Delta y = \frac{\mathcal{B}_y^{\text{t}} - \mathcal{B}_y^{\text{p}}}{\mathcal{B}_h^{\text{t}}}$) between the predicted box and the target box (dashed box). For mAP, we consider seven IoU-thresholds from 0.1 to 0.7. IoU-based distances (AP@50, mAP, IoU) degrade to zero very quickly and prefer axis-aligned boxes, while RoDeO loc degrades more smoothly and is isotropic in Eucledian space.

where $\mathcal{Y}^{\text{t}} = \{y_{\mathcal{B}^{\text{t}}}\}, \mathcal{Y}^{\text{p}} = \{y_{\mathcal{B}^{\text{p}}}\}$ $\quad \forall (\mathcal{B}^{\text{t}}, \mathcal{B}^{\text{p}}) \in \mathcal{M}$, and $y_{\mathcal{B}^{\text{t}}}$ and $y_{\mathcal{B}^{\text{p}}}$ are the classes of the target and predicted box respectively. In contrast to other classification metrics like F1-score or accuracy, MCC only gives positive scores for performance above random, regardless of possible class imbalances in the dataset. Since scores below 0 indicate negative correlations, they should be valued the same as the outputs of completely random models.

**Overprediction, Underprediction** If over- and underpredictions (i.e. False Positives and False Negatives) are not penalized in the sub-metrics, these could be tricked by a large number of predictions. However, they can not be attributed to any of the aforementioned error types alone. Instead, RoDeO integrates them globally into every sub-metric as a linear combination of the score achieved by the sub-metric and zero, weighted by the number of matched over- and underpredictions:

$$\text{RoDeO}^{\text{sub}} = \frac{|\mathcal{M}|}{|\mathcal{M}| + |\mathcal{U}^{\text{t}}| + |\mathcal{U}^{\text{p}}|} \text{RoDeO}^{\text{sub}}_{\text{matched}} \quad \forall \, \text{sub} \in \{\text{loc}, \text{shape}, \text{cls}\} . \qquad (5)$$

**Summary Metric** We further propose a summarization of all the sub-metrics into a single number. We chose the harmonic mean (c.f. Appendix C) for aggregation because a low score in any sub-metric indicates a faulty detection algorithm that should not receive a high score in the summary metric. *RoDeO total*, abbreviated *RoDeO*, is thus computed as

$$\text{RoDeO} = 3 \left( \frac{1}{\text{RoDeO}^{\text{loc}}} + \frac{1}{\text{RoDeO}^{\text{shape}}} + \frac{1}{\text{RoDeO}^{\text{cls}}} \right)^{-1} . \qquad (6)$$

Nevertheless, we recommend using the submetrics as well as the summary metric when measuring performance and comparing algorithms. The summary metric alone does not give a full picture of the different sources of errors.

### 3.1. RoDeO per Class

The computation of RoDeO on a per-class basis is analogous to above. However, for a class $c$, the set of matched boxes only contains the box pairs where the target box belongs to $c$. The unmatched predicted and target boxes, are also filtered to the ones belonging to $c$.

## 4. Experimental Setup

**Dataset**    We perform experiments on the ChestX-ray8 dataset by Wang et al. (2017). It contains $108\,948$ anterioposterior chest radiographs of $32\,717$ unique patients acquired at the National Institutes of Health Clinical Center in the USA. The images are labeled for 8 different pathologies, each image can exhibit multiple pathologies. Additionally, a board-certified radiologist manually labeled 882 of these images with bounding boxes. We used 50% of the images with bounding boxes for evaluation and reserved the other 50% for testing. The rest of the images were used for weakly-supervised training of a machine learning model. There was no patient overlap between all subsets.

**Compared Metrics**    We compare RoDeO to different threshold-based metrics: AP@IoU (the standard object detection metric for natural images) and acc@IoU (Wang et al., 2017).

**Oracle Models and Prediction Corruption**    We study the sensitivity of object detection metrics, including RoDeO, to prediction quality changes and randomness using oracle models. These oracle models are not trained, but instead have access to the target boxes (i.e. the oracle) during inference. The correct bounding boxes are then modified using parameterized random corruption models before they are returned as predictions. We run these oracle models on the test set and study the effect of different corruptions and their parameters. For each corruption setting (with specified parameters), we report the mean results over five runs. We refer to Appendix A for details on the corruption models.

**CheXNet**    To test the usefulness of our proposed metric in practice, we train a CheXNet (Rajpurkar et al., 2017) as a weakly-supervised object detection model on the training dataset, using only image-level labels. For the generation of bounding boxes from heatmaps, we follow Wang et al. (2017). Implementation details are provided in Appendix B.

## 5. Experiments

In this section, we provide examples of undesired behavior of the compared metrics. In all cases, RoDeO behaves as expected. Further experimental results are shown in Appendix D.

**AP@IoU declines sharply with small localization errors.**    Figure 3 shows the influence of position- and size-corruptions on all compared metrics. While AP@IoU declines rapidly with position errors, it stays high under severe size corruptions. acc@IoU shows similar behavior but remains at a high base-value along all corruptions. RoDeO declines more gracefully along both axes and shows a clear separation of the two corruptions.

**Acc@IoU achieves high scores with underprediction.**    In Figure 4, acc@50 increases with higher underprediction probability (i.e. more missed boxes). This leads to higher scores

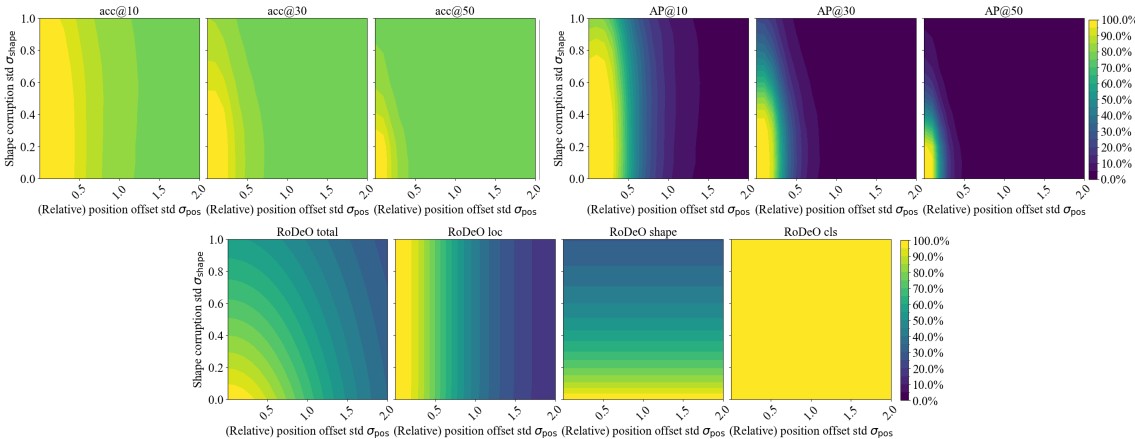

Figure 3: RoDeO (bottom) declines more smoothly than acc@IoU (top left) and AP@IoU (top right) when position and shape are corrupted in our box oracle model. Position and shape corruptions are idenfitied independently by the RoDeO submetrics.

for models that predict no boxes at all, especially at higher IoUs. With more underprediction, a higher number of false positives are exchanged with true negatives, leading to better accuracy. AP@IoU and RoDeO do not suffer from this behavior.

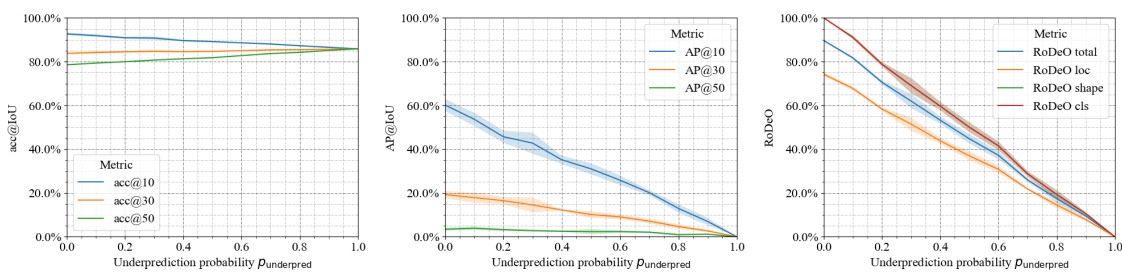

Figure 4: Acc@IoU (left) increases with fewer predictions, whereas AP@IoU (middle) and RoDeO (right) decline as expected when corrupting the box positions ($\sigma_{\mathrm{pos}} = 0.5$) and randomly dropping predicted boxes ($p_{\mathrm{underpred}}$) in our box oracle model.

**AP@IoU ignores overprediction at higher thresholds.** While overprediction leads to an exponential decay (as expected) of all sub-metrics of RoDeO, AP@IoU does not decay with more overprediction at IoUs > 10% in Figure 5. Even worse, AP@50 shows a small but significant increase as more boxes get predicted.

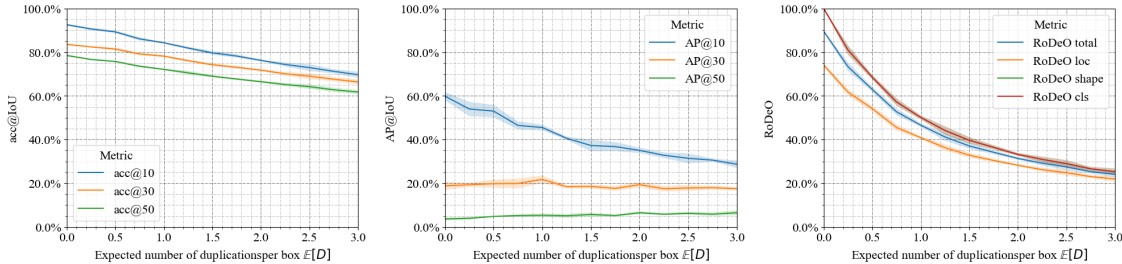

Figure 5: AP@IoU (middle) at higher thresholds increases with more predictions. acc@IoU (left) and RoDeO (right) decline as expected when corrupting the box positions ($\sigma_{\text{pos}} = 0.5$) and randomly predicting multiple boxes per target ($p_{\text{overpred}}$).

Table 1: Qualitative RoDeO, acc@30, and mAP results of a trained CheXNet model. For mAP, we consider seven thresholds from 0.1 to 0.7.

|              | RoDeO/cls | RoDeO/loc | RoDeO/shape | RoDeO | acc@30 | mAP  |
|--------------|-----------|-----------|-------------|-------|--------|------|
| Atelectasis  | 21.6      | 17.3      | 6.9         | 28.4  | 37.1   | 2.0  |
| Cardiomegaly | 55.7      | 70.4      | 45.8        | 55.6  | 86.2   | 31.7 |
| Effusion     | 21.9      | 15.5      | 9.2         | 13.7  | 33.4   | 5.1  |
| Infiltration | 11.0      | 15.9      | 8.1         | 10.8  | 31.6   | 3.0  |
| Mass         | 10.6      | 11.4      | 4.3         | 7.2   | 59.4   | 1.6  |
| Nodule       | 16.9      | 3.0       | 0.8         | 1.9   | 55.7   | 0.0  |
| Pneumonia    | 9.2       | 38.3      | 19.2        | 16.0  | 71.6   | 2.1  |
| Pneumothorax | 26.6      | 14.3      | 12.5        | 16.0  | 65.8   | 1.0  |
| Total        | 19.8      | 19.9      | 10.9        | 15.6  | 55.1   | 5.8  |

### 5.1. Evaluating a Machine Learning Model

Table 1 shows qualitative results of RoDeO and mAP for a trained CheXNet (Rajpurkar et al., 2017). In mAP, atelectasis and pneumonia achieve similar scores, the results of RoDeO paint a more balanced picture. While atelectasis is classified quite well, its predicted shape does often mismatch. For pneumonia, on the other hand, location and shape are predicted quite well, but it was rarely classified correctly by the model – acc@IoU, however, gives one of the highest scores to pneumonia, which is due to the dominance of true negatives in accuracy and renders the metric almost completely useless. RoDeO also reveals that the most severe problem in nodule detection is the shape (or size) mismatch followed by localization, while classification is comparably good.

## 6. Discussion

The above experiments have shown that compared to IoU-based metrics (cf. Sec. 1), RoDeO has more desirable behaviors for medical object detection: It can not be tricked by over- or

under-prediction, it does not consider boxes with class confusion as useless but values the detection of pathologies separately, and it uses a notion of distance that reflects the clinical need for coarse localization better. Our experiments have further shown that different error sources affect sub-metrics independently and that RoDeO degrades gracefully along all sub-metrics. Further evaluation in Appendix D shows that RoDeO also behaves as desired in a plethora of different corruption settings. While we have shown the usefulness of RoDeO for pathology detection, we argue that other medical object detection tasks, such as organ detection, could also benefit from the adavantages of RoDeO.

## 7. Conclusion

We have proposed RoDeO, a novel metric to evaluate object detection algorithms, that is tailored to the requirements in medical pathology detection, especially from chest X-rays. Extensive experiments have shown that RoDeO considers different error-sources more appropriately than previous metrics. It is easy to interpret and use; with robust default values that can easily be adapted to the specific needs of an application. RoDeO allows for better evaluation of pathology detection algorithms and can increase their clinical usefulness.

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

## Appendix A. Corruption Models

**Position and shape corruption**  For position and shape corruption we consider each oracle box individually. Assume the oracle box is defined by its center coordinates $(x, y)$ and its size $(w, h)$. Given the standard deviation $\sigma_{\mathrm{pos}} \in [0, \infty)$ of the (relative) position offset (a parameter of the corruption model which will be scaled by the box size), we sample the corrupted center coordinates as

$$\hat{x} \sim \mathcal{N}\left(x, (w \cdot \sigma_{\mathrm{pos}})^2\right), \qquad\qquad \hat{y} \sim \mathcal{N}\left(y, (h \cdot \sigma_{\mathrm{pos}})^2\right), \qquad (7)$$

where for $\sigma_{\mathrm{pos}} = 0$ we define $\hat{x} = x$ and $\hat{y} = y$. This position corruption model is used in Figures 3, 4, and 5 from the main paper and Figures 8, 10, and 11 in Appendix D.

For Figure 8 in Appendix D, we additionally corrupt the box positions by introducing a position bias $|\Delta \boldsymbol{p}| \in [0, \infty)$. For each predicted box, we randomly sample an angle $\phi \in [0, 2\pi]$ and offset the box into that direction, i.e.

$$\hat{x} = x + \frac{|\Delta \boldsymbol{p}| \cdot \cos\phi}{w}, \qquad\qquad \hat{y} = y + \frac{|\Delta \boldsymbol{p}| \cdot \sin\phi}{h}. \qquad (8)$$

For shape corruption, we assume a multiplicative corruption model and therefore sample from the Lognormal distribution as follows

$$\ln(\hat{w}) \sim \mathcal{N}\left(\ln(w), \sigma_{\mathrm{shape}}^2\right) \qquad\qquad \ln(\hat{h}) \sim \mathcal{N}\left(\ln(h), \sigma_{\mathrm{shape}}^2\right), \qquad (9)$$

where for $\sigma_{\mathrm{shape}} = 0$, we define $\hat{w} = w$ and $\hat{h} = h$. This shape corruption model is used in Figure 3 in the main paper.

For Figure 9 in Appendix D, we corrupt size and aspect ratio of boxes independently. The size is again corrupted multiplicatively, but width and height are corrupted by the same factor $s$ sampled from a Lognormal as

$$\ln(\Delta s) \sim \mathcal{N}\left(\ln(1), \sigma_{\mathrm{size}}^2\right), \qquad (10)$$

and then new width and height are computed as

$$\hat{w} = \Delta s \cdot w, \qquad\qquad \hat{h} = \Delta s \cdot h. \qquad (11)$$

For corrupting the aspect ratio in Figure 9 the area $A = w \cdot h$ of a box is kept constant while modifying the aspect ratio $a = \frac{w}{h}$ as

$$\ln(\hat{a}) \sim \mathcal{N}\left(\ln(a), \sigma_{\mathrm{ratio}}^2\right), \qquad (12)$$

and then computing the new width and height as

$$\hat{w} = \sqrt{A \cdot \hat{a}}, \qquad\qquad \hat{h} = \sqrt{\frac{A}{\hat{a}}}. \qquad (13)$$

**Class Underprediction**  For class underprediction, we consider each sample independently and randomly decide for each positive class $c^+$ in its oracle whether to flip it to negative by sampling from a Bernoulli distribution with probability $p_{\mathrm{underpred}} \in [0, 1]$. We then discard all boxes for the flipped classes in the current sample. For simulating more realistic behavior we additionally apply random position corruption with $\sigma_{\mathrm{pos}} = 0.5$ to all remaining boxes. This corruption model is used in Figure 4 from the main paper.

**Class Overprediction**   For class overprediction we proceed similarly and randomly decide for each negative class $c^-$ in its oracle whether to flip it to positive by sampling from a Bernoulli distribution with probability $p_{\text{overpred}} \in [0, 1]$. We then generate one additional box for each of the classes that have been flipped to positive, where the box size is set to 0.25 for width and height (the mean box size in the dataset) and the center position is sampled uniformly such that the whole box is contained within the image. This corruption model is used in Figure 10 in Appendix D.

**Box Overprediction**   For box overprediction per class we do not predict additional classes but duplicate oracle boxes. We, therefore, sample the number of duplications $D$ of each box independently from a geometric distribution with a specified expected number of duplications $\mathbb{E}[D]$. Just like in the underprediction scenario, we additionally apply random position corruption with $\sigma_{\text{pos}} = 0.5$ to the original and all duplicated boxes, resulting in duplicated boxes slightly deviating from the original boxes. This corruption model is used in Figure 5 from the main paper.

**Class confusion**   For class confusion, we consider each sample individually and randomly decide which of the classes $c \in \mathcal{C}$ to confuse with each other. We, therefore, first randomly select a subset of classes to confuse by sampling from a Bernoulli distribution with probability $p_{\text{cls-confuse}} \in [0, 1]$ for each class independently, where $p_{\text{cls-confuse}} = 0$ corresponds to no confusion at all and $p_{\text{cls-confuse}} = 1$ to confusing all classes. We then randomly permute the set of selected classes in this sample while leaving the non-selected classes unchanged. This corruption model is used in Figure 11 in Appendix D.

**Class oracle with random positioned boxes**   We also experiment with randomly positioning boxes. In this case we assume only a class oracle, i.e. we know the correct set of positive classes but not the related bounding boxes. First, we randomly add additional classes (i.e. class overprediction) by randomly flipping negative classes to positive using a Bernoulli distribution with probability $p_{\text{overpred}}$. Based on this new set of positive classes (the correctly positive and the flipped negative classes) we sample one bounding box per positive class. The sampled boxes are square with pre-defined size $s$ and each box position is sampled uniformly such that the whole box is contained within the image. This corruption model is used in Figure 7 in Appendix D.

## Appendix B.  CheXNet Implementation Details

Here we describe the implementation details of the CheXNet model (Rajpurkar et al., 2017). It was trained using Adam (Kingma and Ba, 2014) with a learning rate of $3.6e - 5$, weight decay of $1e - 6$, and gradient clipping at norm 1.0. Both training and testing images were resized to $224 \times 224$ pixels, and normalized to zero mean and unit variance using the statistics of the training dataset. During training, we randomly applied random color jitter and random gaussian blurring to the images, each with a probability of 50%. No data augmentation was applied during testing. We trained for a maximum number of 50000 iterations with early stopping (patience $= 10000$) and a batch size of 128 on a single Nvidia RTX A6000 GPU.

**Bounding Box Generation from Heatmaps** We consider an input image represented as $x \in \mathbb{R}^{H \times W}$. A model produces $C$ heatmaps $\hat{s} \in \mathbb{R}^{H \times W}$, one for each class. These heatmaps are generated by upsampling the spatially reduced score maps $s \in \mathbb{R}^{h \times w}$ per class using a bilinear interpolation. Following the box proposal method described in (Wang et al., 2017), we normalize the heatmaps per image between 0 and 255 and create two sets of binarized maps by threshold them at 60 and 180. From the resulting binarized heatmaps, box proposals are drawn around each connected component, denoted as $\hat{s}k$. The predicted class for each box is determined by the connected component it encloses.

## Appendix C. Additional Discussion of Design Decisions

**Shape Sub-Metric using cIoU** Fig 6 compares the cIoU with the Hausdorff similarity when evaluating shape similarities (size and aspect-ratio) between bounding boxes. While the Hausdorff similarity decreases linearly and has no lower bound, cIoU decreases faster for small differences while decreasing slower for larger differences until converging towards zero. Additionally, cIoU is measured relative to the target box size which is a desired property. Lastly, cIoU is bound between zero and one, making it an easily interpretible sub-metric.

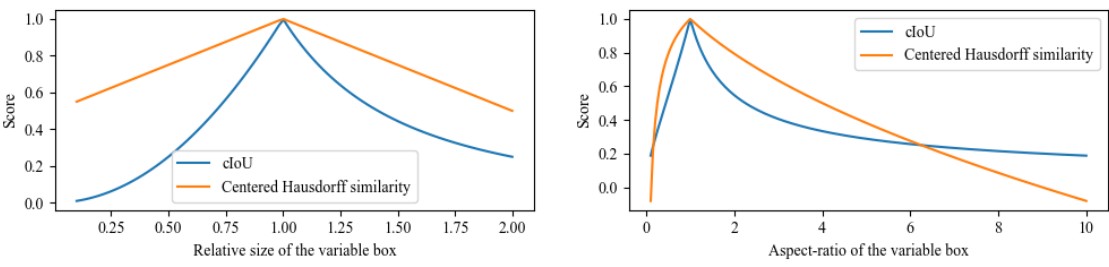

Figure 6: cIoU and Hausdorff similarity of a fixed box and another box with varying size with fixed aspect-ratio (a) or varying aspect-ratio with fixed size (b).

**Summary Metric using Harmonic Mean** We propose the summary metric to describe the detection quality of a model in a single number. Therefore, we impose the following requirements on it: (i) the maximum value of the summary metric is capped based on the value of the worst sub-metric, i.e. if one sub-metric performs low, the summary metric cannot be arbitrarily improved by increasing the other sub-metrics; (ii) improving any of the sub-metrics should lead to an improved summary metric, such that any form of improvement is respected in the summary metric.

The harmonic mean fulfills both of these requirements, except if a sub-metric is exactly zero, in which case the summary metric is zero as well, therefore breaking requirement (ii). However, this is only possible for the classification sub-metric, in which case we argue that the model does not predict any meaningful boxes and reporting zero for the summary metric seems sensible. Other possible choices include the arithmetic mean, which however

does not fulfill requirement (i), or the minimum of all sub-metrics, which however does not fulfill requirement (ii).

## Appendix D. Additional Results

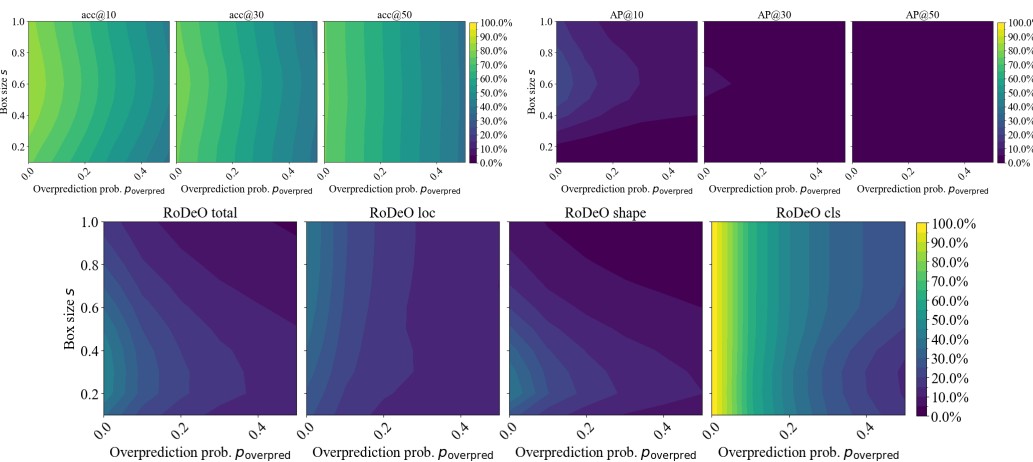

Figure 7: Class oracle with random positioned boxes with class oversampling. RoDeO (ours) and AP@IoU score low, as expected for randomly positioned boxes, while acc@IoU reports high values. RoDeO shape achieves its maximum value at the expected box size in the dataset (roughly 0.2) while AP@IoU and acc@IoU achieve better scores with boxes larger than the expected size.

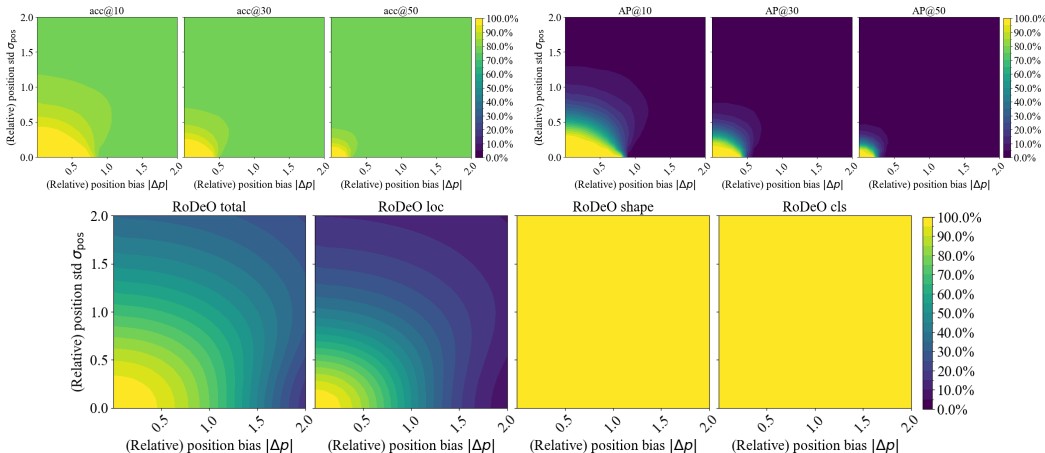

Figure 8: Box oracle with position offsets. AP@IoU and acc@IoU decline sharply with small localization errors, especially when the predictions are consistently offset (position bias), while RoDeO (ours) declines more smoothly for both, increasing position bias and std. RoDeO shape and RoDeO cls are invariant to position offsets, as expected.

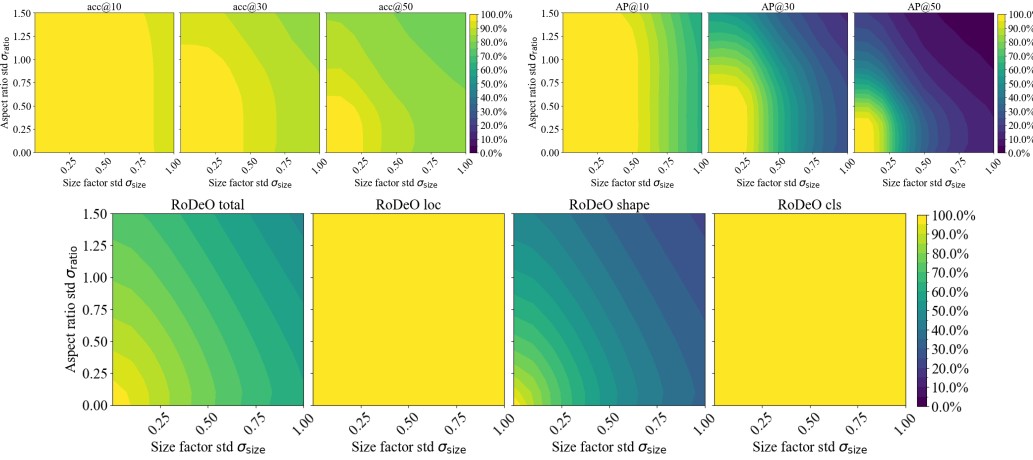

Figure 9: Box oracle with random shape corruptions. Acc@IoU is mostly invariant to all shape corruptions, while AP@10 is mostly invariant to aspect ratio corruptions. RoDeO (ours) on the other hand declines smoothly for both, aspect ratio and size, corruptions, while RoDeO loc and RoDeO cls are invariant to these corruptions, as expected.

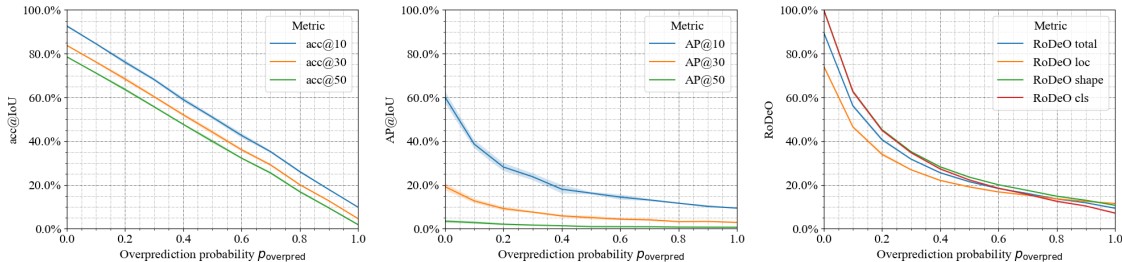

Figure 10: Box oracle with $\sigma_{\text{pos}} = 0.5$ (relative) position variation and oversampled classes. All three metrics decline smoothly with increasing overprediction. While AP@IoU and RoDeO show an exponential decay, as expected when sampling overprediction from a geometric distribution, acc@IoU declines linearly.

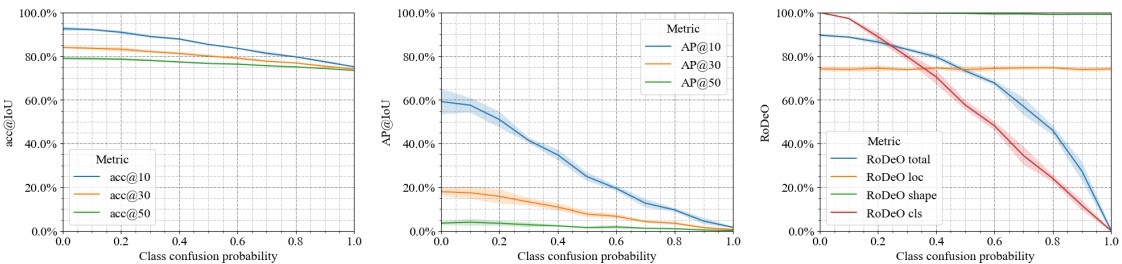

Figure 11: Box oracle with $\sigma_{\text{pos}} = 0.5$ (relative) position variation and random class swaps. All three metrics decline smoothly when classes are mispredicted, acc@IoU however never reaches zero even when classes are completely mispredicted. RoDeO loc and RoDeO shape are invariant to these class corruptions, as expected.

## Appendix E. Definitions

**Matthews Correlation Coefficient (MCC)**  The *Matthews Correlation Coefficient (MCC)* (Matthews, 1975) is defined as follows:

$$\text{MCC} = \frac{\text{TP} * \text{TN} - \text{FP} * \text{FN}}{\sqrt{(\text{TP} + \text{FP})(\text{TP} + \text{FN})(\text{TN} + \text{FP})(\text{TN} + \text{FN})}}, \tag{14}$$

where TP are the true positives, FP are the false positive, TN are the true negatives, and FN are the false negatives of a classification model over the whole evaluation dataset. MCC is bound between $-1$ and 1, where a perfect classification is 1, a random model achieves 0 and negative values indicate negative correlations.

**Intersection over Union (IoU)**  The *Intersection over Union (IoU)* between two bounding boxes $\mathcal{B}^{\mathfrak{t}}$ and $\mathcal{B}^{\mathfrak{p}}$ is defined as follows:

$$\text{IoU} = \frac{|\mathcal{B}^{\mathfrak{t}} \cap \mathcal{B}^{\mathfrak{p}}|}{|\mathcal{B}^{\mathfrak{t}} \cup \mathcal{B}^{\mathfrak{p}}|} = \frac{|\mathcal{B}^{\mathfrak{t}} \cap \mathcal{B}^{\mathfrak{p}}|}{\mathcal{B}_w^{\mathfrak{t}} \mathcal{B}_h^{\mathfrak{t}} + \mathcal{B}_w^{\mathfrak{p}} \mathcal{B}_h^{\mathfrak{p}} - |\mathcal{B}^{\mathfrak{t}} \cap \mathcal{B}^{\mathfrak{p}}|}, \tag{15}$$

where

$$|\mathcal{B}^{\mathfrak{t}} \cap \mathcal{B}^{\mathfrak{p}}| = (\min(\mathcal{B}_{x+}^{\mathfrak{t}}, \mathcal{B}_{x+}^{\mathfrak{p}}) - \max(\mathcal{B}_{x-}^{\mathfrak{t}}, \mathcal{B}_{x-}^{\mathfrak{p}}))(\min(\mathcal{B}_{y+}^{\mathfrak{t}}, \mathcal{B}_{y+}^{\mathfrak{p}}) - \max(\mathcal{B}_{y-}^{\mathfrak{t}}, \mathcal{B}_{y-}^{\mathfrak{p}})) \tag{16}$$

with $\mathcal{B}_{x-} = \mathcal{B}_x - \frac{\mathcal{B}_w}{2}$ and $\mathcal{B}_{y-} = \mathcal{B}_y - \frac{\mathcal{B}_h}{2}$ are the upper left coordinates of each box, and $\mathcal{B}_{x+} = \mathcal{B}_x + \frac{\mathcal{B}_w}{2}$ and $\mathcal{B}_{y+} = \mathcal{B}_y + \frac{\mathcal{B}_h}{2}$ are the lower right coordinates.

**Centered IoU (cIoU)**  The *centered IoU (cIoU)* is computed as the normal IoU, but assuming the same position for both boxes $\mathcal{B}^{\mathfrak{t}}$ and $\mathcal{B}^{\mathfrak{p}}$, i.e. $\mathcal{B}_x^{\mathfrak{t}} = \mathcal{B}_x^{\mathfrak{p}}$ and $\mathcal{B}_y^{\mathfrak{t}} = \mathcal{B}_y^{\mathfrak{p}}$.

**Generalized IoU (gIoU)**  The *generalized IoU (gIoU)* (Rezatofighi et al., 2019) is the IoU minus an additional term which is the ration between the convex hull $\mathcal{C}$ of the predicted and target box minus their Union and $\mathcal{C}$.

$$\text{gIoU} = \text{IoU} - \frac{|\mathcal{C}| - |\mathcal{B}^{\mathfrak{t}} \cup \mathcal{B}^{\mathfrak{p}}|}{|\mathcal{C}|}, \tag{17}$$

where the convex hull $|\mathcal{C}| = (\max(\mathcal{B}_{x+}^{\mathfrak{t}}, \mathcal{B}_{x+}^{\mathfrak{p}}) - \min(\mathcal{B}_{x-}^{\mathfrak{t}}, \mathcal{B}_{x-}^{\mathfrak{p}}))(\max(\mathcal{B}_{y+}^{\mathfrak{t}}, \mathcal{B}_{y+}^{\mathfrak{p}}) - \min(\mathcal{B}_{y-}^{\mathfrak{t}}, \mathcal{B}_{y-}^{\mathfrak{p}}))$.

**Computing IoU-based Classification Metrics for Object Detection**  For each image, there is a set of predicted ($\mathcal{M}^{\mathfrak{p}}$) and target boxes ($\mathcal{M}^{\mathfrak{t}}$).

1. **True positives** are predicted boxes that have the same class as a target box, and an IoU larger than or equal to a predefined threshold $t$ ($\text{IoU}(\mathcal{B}^{\mathfrak{p}}, \mathcal{B}^{\mathfrak{t}}) \leq t \wedge y_{\mathcal{B}^{\mathfrak{p}}} = y_{\mathcal{B}^{\mathfrak{t}}}$). Note that there is a one-to-one mapping between predicted and target boxes: every predicted box can only be mapped to one target box and every target box only to one predicted box

2. If for a target box there are more than one predicted boxes that satisfy the above constraints, only the one with the maximum IoU can be counted as true positive, all others are **false positives**. Otherwise there could be more true positives than actual targets which would lead to recall larger than 1

3. All predicted boxes with IoU smaller than $t$ to any target box are also considered **false positive**

4. If for a target box no predicted box satisfies the constraints in 1., it is considered a **false negative**

5. A **true negative** is when an image has no predicted and no target boxes of a specific class

From here, typical classification metrics such as accuracy@IoU, precision@IoU, recall@IoU, or F1@IoU can be computed. For AP@IoU (and mAP), each predicted box further needs a confidence score to integrate over. At every point on the curve, only boxes with a confidence threshold larger than or equal to the current confidence threshold are considered to be in $\mathcal{M}^{\mathfrak{p}}$ for the above calculations.

