# OpenReview forum: "Robust Detection Outcome: A Metric for Pathology Detection in Medical Images"
_MIDL.io/2023/Conference — MIDL 2023 Poster_

### Official Review · Reviewer_usgB · 2023-02-06

**Confidence:** 4
**Preliminary Rating:** 4
**Recommendation:** Oral

**Summary:**

The authors introduced Robust Detection Outcome (RoDeO), a new metric for pathology detection based on three types of errors that cause the detection quality to deteriorate: classification errors, localization errors, and shape mismatch. Its quite interesting. However, the experiments were limited to be generalized in view point of developing metrics.


**Strengths:**

Robust Detection Outcome (RoDeO), a new metric for pathology detection based on three types of errors that cause the detection quality to deteriorate: classification errors, localization errors, and shape mismatch. Its quite interesting Clinically unmet needs based new detection evaluation metrics were suggested.

**Weaknesses:**

The experiments were limited to be generalized in view point of developing metrics. It should be evaluated in various kinds of different datasets. In general viewpoint, this evaluation metrics also can be applicable to natural image detection problem

**Deanonymize Review:**

yes

**Detailed Comments:**

well writen.

**Paper Type:**

methodological development

**Questions To Address In The Rebuttal:**

The experiments were limited to be generalized in view point of developing metrics. It should be evaluated in various kinds of different datasets. In general viewpoint, this evaluation metrics also can be applicable to natural image detection problem

---

### Official Review · Reviewer_nEXM · 2023-02-06

**Confidence:** 3
**Preliminary Rating:** 2

**Summary:**

After addressing the weaknesses of often used evaluation metrics for pathology detection, the authors propose RoDeO, a metric designed for this exact problem. Each component of RoDeO is introduced to improve over different issues with alternative metrics, such as over-, underprediction, shape similarities, and a meaningful aggregation of the sub-metrics to yield a single value. The metric is shown to be robust and outperform the other methods.

**Strengths:**

The authors propose a robust and explainable evaluation metric for object detection using a combination of three sub-metrics that provide a better insight into the behavior of trained machine learning models (or other detection algorithms) than single evaluation metrics. Both the problem of object detection and machine learning methods can benefit from more issue-specific explainability, therefore the manuscript certainly shows scientific merit.

The authors also perform a thorough assessment of the alternative metrics and exploit their weaknesses to build RoDeO which, by design, becomes more useful in extreme scenarios than the metrics to compare to.

**Weaknesses:**

Although I believe that a mindfully constructed evaluation metric is certainly useful, the selected evaluation metrics for comparison are quite arbitrary. As an example, a quick glance at the confusion matrix makes it very clear that neither the average precision metric nor the accuracy will be able to handle over- and underprediction, however metrics such as precision and recall could paint a more detailed picture. These metrics are briefly mentioned, however they are not included in the evaluations. I believe a more concise collection of baseline metrics would lead to a more interesting evaluation of RoDeO, further strengthening the paper.

**Deanonymize Review:**

no

**Detailed Comments:**

- Page 2 row 6 introduces the IoU abbreviation, however it has been used before.
- The labels for the colormaps on Figure 3 are cropped. In the appendix, similar figures look OK.
- "Since these are typically hard to obtain in pathology detection, and earlier works have used significantly lower thresholds". This sentence from page 2 is difficult to follow, consider rephrasing it. If I understood the meaning correctly, perhaps you should remove the "since".
- I believe the impact of the work and of RoDeO would benefit largely from providing the source code to its implementation.

**Paper Type:**

methodological development

**Questions To Address In The Rebuttal:**

- Assessing Figure 1., it seems like the simple IoU metric is a good enough evaluation metric if it was multiplied with a similar indicator function as in Equation 1. to remove unmatching classes. This would give a more fair comparison to RoDeO. Arguably the scenario presented in the left figure would not be reported as an IoU of 81.5% in a scientific paper, but 0% as the classes don't match. Please explain why this over-simplified version of the IoU was used to show the drawbacks of simpler evaluation metrics, if only slight adjustments could improve the methods significantly.
- The authors should justify the baseline metrics they selected for comparison to RoDeO. Very popular evaluation metrics (precision, recall, F1 score) are mentioned but not used for comparison, and other popular metrics, such as MCC and DICE coefficient (although MCC is used in RoDeO) are not even considered as alternatives. The metrics selected for comparison are also by definition not able to perform well for some of the evaluations they are exposed to (eg. the sharp decline of AP for localization errors, and using IoU without an indicator function used in Equation 1.)
- The authors should also detail how they have decided on the specific sub-components of RoDeO and how the components have been optimized. For example why they have chosen cIoU as a shape similarity instead of other distance metrics, and if summary metrics other than the harmonic average were explored or not.
- As a specific example, I would be curious to see how the harmonic mean of three simpler metrics such as acc@50, recall@50 and precision@50 would fare against RoDeO, as based on Figure 1 it should be able to handle all four cases, given that the previously mentioned indicator function from Equation 1 is applied on IoU.

---

### Official Review · Reviewer_GnTs · 2023-02-07

**Confidence:** 4
**Preliminary Rating:** 5
**Recommendation:** Oral

**Summary:**

The common metrics such as IoU, mAP, etc., that are used for evaluation of object detection and localization in natural images are inadequate for use in a clinical setting. The authors propose a more insightful metric, RoDeO, for evaluation of pathology detection in medical imaging. RoDeO accounts for the shape, location, and the class of pathology detected in the image and provides a summarized score(authors recommend using the sub-metrics along with the summary score to grasp the full picture). In this paper, RoDeO metrics have been obtained along with the baseline metrics on a large dataset to establish it's usefulness.

**Strengths:**

The paper is written very well. The motivation is clearly established.

The writing is such that the shortcomings of the metrics currently used in pathology detection and the method to address them with the new proposed metric are easy to comprehend.

The experimental setup and the evaluation of the machine learning model are convincing.

**Weaknesses:**

I really like the way this paper has been written and I do not see any major weakness to mention. It is quite convincing that the proposed metric can provide a more insightful evaluation of pathology detection models.

**Deanonymize Review:**

no

**Paper Type:**

both

**Questions To Address In The Rebuttal:**

The paper is well written with all the appropriate citations.

It is recommended that RoDeO summary metric not be used by itself, but in combination with the sub-metrics for better insight. How can RoDeO summary metric misinform the evaluation of detection models?

---

### Meta-Review · Area_Chair_aDH8 · 2023-02-22

**Recommendation:** Accept (Poster)
**Confidence:** 3

**Metareview:**

The paper proposes a metric to better reflect pathology detection in medical images than the IoU metric used in computer vision, with chest x-rays as a specific application.

The reviewers agree on the motivation and clarity of the paper. One of the reviewers has several questions about inner workings of the metric, and there is a lively discussion, which we thank the authors and the reviewers for.

I am somewhat surprised that the paper is motivated by specific clinical requirements of pathology detection, but is mostly based in computer vision work, with only few sources from medical imaging or clinical venues. There have been a lot of efforts in the community to address appropriate metric choice, see https://arxiv.org/pdf/2206.01653.pdf . However, despite this omission I think there is value in how the validation is done, which could be also relevant for other metrics in used, so I think it would be worth discussing the paper at the conference.